# GWAS and Meta-QTL Analysis of Kernel Quality-Related Traits in Maize

**DOI:** 10.3390/plants13192730

**Published:** 2024-09-29

**Authors:** Rui Tang, Zelong Zhuang, Jianwen Bian, Zhenping Ren, Wanling Ta, Yunling Peng

**Affiliations:** 1College of Agronomy, Gansu Agricultural University, Lanzhou 730070, China; tangr7450@163.com (R.T.); zhuangzl3314@gmail.com (Z.Z.); bjwen1018@163.com (J.B.); renzp1003@163.com (Z.R.); kellytwl@163.com (W.T.); 2Gansu Provincial Key Laboratory of Aridland Crop Science, Gansu Agricultural University, Lanzhou 730070, China

**Keywords:** maize, quality traits, GWAS, candidate genes, meta-QTL

## Abstract

The quality of corn kernels is crucial for their nutritional value, making the enhancement of kernel quality a primary objective of contemporary corn breeding efforts. This study utilized 260 corn inbred lines as research materials and assessed three traits associated with grain quality. A genome-wide association study (GWAS) was conducted using the best linear unbiased estimator (BLUE) for quality traits, resulting in the identification of 23 significant single nucleotide polymorphisms (SNPs). Additionally, nine genes associated with grain quality traits were identified through gene function annotation and prediction. Furthermore, a total of 697 quantitative trait loci (QTL) related to quality traits were compiled from 27 documents, followed by a meta-QTL analysis that revealed 40 meta-QTL associated with these traits. Among these, 19 functional genes and reported candidate genes related to quality traits were detected. Three significant SNPs identified by GWAS were located within the intervals of these QTL, while the remaining eight significant SNPs were situated within 2 Mb of the QTL. In summary, the findings of this study provide a theoretical framework for analyzing the genetic basis of corn grain quality-related traits and for enhancing corn quality.

## 1. Introduction

Maize (*Zea mays* L.) is a vital source of food, livestock feed, bioenergy, and industrial raw materials [1]. Maize grains are rich in essential nutrients, including starch, protein, oil, water-soluble polysaccharides, vitamins, and minerals [2]. Among these components, starch is recognized as a sustainable and cost-effective biodegradable natural polysaccharide, comprising approximately 65–70% of the total endosperm [3]. Proteins are vital for living organisms, fulfilling crucial roles in energy metabolism. At physiological maturity, the storage proteins in maize kernels account for approximately 10% of the grain’s weight. Furthermore, maize grains contain a crude fat content of about 4–6%, making them a valuable raw material for the production of edible oils [4]. With the rapid advancements in animal husbandry and processing industries, the quality of corn kernels has received increasing attention. Currently, many corn varieties exhibit suboptimal nutritional quality, failing to meet essential health requirements. Consequently, exploring genetic enhancements related to corn kernel quality traits is crucial for improving nutritional standards and promoting the development of high-quality corn varieties.

The quality traits of corn kernels primarily encompass protein content, starch content, and oil content. The levels of these nutrients serve as critical indicators of corn quality [5]. In recent years, advancements in modern molecular biology have enabled researchers, both domestically and internationally, to conduct quantitative trait loci (QTL) mapping and genome-wide association studies (GWAS) on the traits associated with corn grain quality. Numerous QTL and candidate genes that are closely linked to quality traits, including grain starch, protein, and oil content, have been identified across various chromosomes. For instance, Mangolin et al. [6] detected 13 QTL in an F_2:3_ population through QTL mapping focused on corn kernel oil content. Additionally, Zhang et al. [7] utilized three populations across three environments to identify 38 QTL associated with corn grain quality traits. Cook et al. [8] performed a correlation analysis on 282 inbred line quality traits, identifying 121 SNP sites related to starch, protein, and fat content, thus marking the first report on the study of quality traits through GWAS. Furthermore, Liu et al. [9] conducted a genome-wide association analysis on the quality traits of 263 corn inbred lines, identifying four loci associated with starch content. Lastly, Li et al. [10] analyzed the genetic structure underlying oil biosynthesis in corn kernels.

While numerous QTL and SNPs linked to quality traits have been reported, variations among studies regarding experimental environments, population maps, population types, trait selection, and statistical methods complicate the identification of QTL for all control traits within a single study. This challenge is further exacerbated by the limitations in the density of QTL mapping markers, which result in considerable genetic distances within the confidence intervals of positioned QTL, thereby diminishing their validity [11]. A meta-analysis, supported by mathematical models, presents a viable strategy to enhance the precision and effectiveness of QTL positioning by amalgamating data from diverse studies to optimize the confidence intervals of QTL [12]. Despite these advantages, the synergy between GWAS and meta-QTL in jointly analyzing corn quality-related traits remains relatively unexplored. The pursuit of high-yield corn breeding necessitates a multifaceted approach, incorporating correlation analyses across distinct populations and environments to identify stable correlation sites and superior alleles. This study investigated a natural population comprising 260 maize inbred lines from various sources, conducting a GWAS investigation related to quality traits to decipher the genetic underpinnings of maize quality traits and identify the SNP sites significantly linked to these traits. By integrating QTL for multiple quality traits, meta-analysis methods were employed to discern “consistent” QTL (meta-QTL) associated with corn quality traits, thereby scouting for candidate genes and providing a foundation for the precise positioning and cloning of pivotal genes linked to corn quality traits, ultimately contributing to the enhancement of corn quality through molecular marker-assisted selective breeding.

## 2. Results

### 2.1. Phenotypic Analysis of Quality-Related Traits

Basic statistical analyses revealed extensive phenotypic continuous variation across all traits, with coefficients of variation ranging from 3% to 18% (Table 1). The three quality traits exhibited a positive skew in their distribution (Appendix A), indicating that these traits are typical quantitative traits, with their inheritance primarily influenced by polygenic factors.

Correlation analysis revealed a highly significant negative correlation between starch content in the grains and both protein and oil contents, while protein and oil contents exhibited a positive correlation (Appendix A). This indicates the presence of interactions, influences, and synergies among the grain traits related to corn quality, which, in turn, regulate and affect the overall quality of corn. Based on the analysis of various phenotypic traits, three quality-related traits were identified as suitable for genome-wide association study (GWAS) analysis.

### 2.2. GWAS Analysis

This study employed mixed linear models (MLM) and the FarmCPU model to perform a genome-wide association study (GWAS) analysis for each trait. The Manhattan plots and QQ plots indicated that false positives were effectively controlled for each trait (Figure 1). In the GWAS analysis results for the three related traits, using a significance threshold of *p* = 1 × 10^−5^, a total of 23 significant SNPs were identified across the three traits in the associated group (Appendix A), comprising 11 SNPs for oil, 5 for starch, and 7 for protein.

### 2.3. Expression Analysis of Candidate Genes

The expression characteristics of the candidate genes identified at the main stages and in various tissues were analyzed using the public database qTeller. The results indicated that 32 out of the 92 candidate genes exhibited expression levels, and these genes were categorized into four groups based on their distinct expression patterns (Figure 2; Appendix A). In the first category, expression levels were high across nearly all tissues and stages. Notably, GRMZM2G089484 (mpk6—MAP kinase 6) exhibited the highest expression level in all tissues, and ZmMAKP6 was implicated in the regulation of corn kernel weight. Additionally, the expression levels of GRMZM2G004182, GRMZM2G063316, and GRMZM2G039922 were consistently high across all stages in nearly all tissues. The second category includes genes that are highly expressed only in specific tissues; for instance, GRMZM2G301122 demonstrated the highest expression level in the endosperm. GRMZM2G136910 and GRMZM2G040102 were highly expressed in the entire seed, endosperm, and silk, suggesting their involvement in the regulation of multiple stress tolerances. Category III genes are expressed in all tissues, albeit at low levels, while Category IV genes are also expressed in all tissues.

### 2.4. Screening and Functional Analysis of Candidate Genes

Candidate genes were identified based on the presence of single nucleotide polymorphisms (SNPs) within the population, alongside a linkage disequilibrium (LD) decay distance of 100 kb (r^2^ = 0.1). The search utilized the B73_RefGen_v3 reference genome, employing a 100 kb interval both upstream and downstream of the SNPs as the candidate region. In total, 36 genes were identified within the oil candidate region, 23 genes within the starch candidate region, and 33 genes within the protein candidate region (Appendix A). Functional annotation of these genes revealed nine potential candidate genes primarily encoding the various proteins and transcripts associated with growth and developmental factors (Table 2).

### 2.5. QTL Distribution of Corn Grain Quality Traits

This study compiled the quantitative trait loci (QTL) associated with three quality-related traits, namely, oil content (OC), starch content (SC), and protein content (PC), from 27 relevant publications. A total of 697 grain quality-related QTL that met the predetermined criteria were identified. These QTL exhibited an uneven distribution across the chromosomes, with counts ranging from 36 to 101. Notably, chromosome 7 harbored the fewest QTL, while chromosome 5 contained the most. Specifically, the number of QTL associated with OC, PC, and SC were 222, 229, and 246, respectively (Appendix A).

### 2.6. Meta-QTL Analysis

The quality-related quantitative trait loci (QTL) for corn, as compiled in this study, were identified by various research groups employing different molecular markers and genetic maps. To extract valuable and comprehensive information from these collected QTL for future research, they were projected onto the published reference map, IBM2 2008 Neighbors. A total of 331 QTL, representing 47.48% of the total QTL gathered from the literature, were successfully projected onto this newly developed high-density reference map (Figure 3 and Appendix A; Appendix A). The remaining QTL could not be projected onto the reference map, which may be attributed to a lack of common markers between the original and reference maps or to the QTL exhibiting low phenotypic variance explained (R^2^), resulting in large confidence intervals (CIs) [13].

Through meta-analysis, we identified a total of 40 QTL associated with quality traits, based on the model with the lowest AIC (Akaike information criterion) value. Each of these QTL encompass between 3 and 22 initial QTL. The distribution of the 40 identified QTL across the 10 maize chromosomes is uneven: 5 QTL are located on chromosome 5, 2 QTL each on chromosomes 3 and 10, 3 QTL each on chromosomes 1, 2, and 9, and 3 QTL each on chromosomes 4 and 7. Additionally, there are 4 QTL on chromosome 6 and 8 QTL on chromosome 8. The confidence intervals (95%) of the detected meta-QTL range from 1.15 to 11.76 cM, with an average of 5.16 cM (Appendix A). By comparing the positions of genetic markers at both ends of the QTL on the B73 genome (AGPv 3), we identified a total of 1639 candidate genes within these QTL regions (Appendix A). Notably, MQL 4 contained the highest number of candidate genes, totaling 158, while MQL 33 had at least 7 candidate genes. Among the 22 QTL, we detected 23 reported functional genes and candidate genes related to corn quality traits, suggesting that utilizing meta-analysis is both effective and feasible for exploring the genes associated with corn quality traits.

The significant SNPs associated with 23 quality traits, as identified through GWAS, were compared with the physical coordinates of 40 QTL. The results indicated that within the intervals of MQTL10, MQTL13, and MQTL39, the SNPs 4_167038832, 9_136920692, and 5_5579448 were respectively situated within the MQTL intervals (Figure 4). Additionally, eight significant SNPs were found outside the QTL intervals but were located very close to them, at a distance of less than 2 Mb. These findings further validate the accuracy of the quality trait-related SNPs identified in this study.

## 3. Discussion

### 3.1. Genetic Basis of Kernel Quality-Related Traits

Maize has emerged as one of the most significant crops for food, livestock feed, and fuel globally. In recent years, the rapid advancement of the animal husbandry and processing industries has led to increased research attention on the quality of corn kernels [14]. Corn kernel quality traits are complex quantitative traits, governed by major effect genes as well as a multitude of micro-effect genes. Current research has focused on quality-related genes, with researchers identifying the Shrunken2 and Opaque2 mutants associated with the protein content of corn kernels. Analysis revealed that while the Opaque2 mutant decreased gliadin content, it simultaneously increased non-gliadin content, resulting in an overall increase in total protein content [15,16,17,18]. Hu et al. [19] utilized a multi-parent population to conduct a localization analysis of corn grain starch content, identifying a candidate gene, ZmTPS9, which encodes trehalose-6-phosphate synthase. Knocking out the ZmTPS9 gene can enhance the starch content of corn grains, thereby increasing their weight. This finding suggests that the ZmTPS9 gene plays a dual role in regulating corn grain starch synthesis and grain development. Additionally, Shen et al. [20] investigated the regulatory mechanisms of the overexpressed genes ZmLEC1 and ZmWR11 on the oil content of corn kernels. Their results indicated that the overexpression of ZmLEC1 could increase the oil content of corn kernels by 48.7%, whereas the overexpression of ZmWR11 resulted in a more moderate increase. Importantly, increasing the oil content of corn kernels does not lead to a decrease in yield. The primary chemical components of corn kernels include protein, starch, and oil. In corn kernels, 70% of the protein is located in the germ, 98% of the starch is found in the endosperm, and 85% of the oil is present in the embryo. This study examines various quality traits in natural populations, all of which exhibit a normal or approximately normal distribution. Notably, the starch content of corn kernels is highly negatively correlated with both kernel protein and oil contents, while kernel protein content is positively correlated with oil content [21,22,23]. The observed positive correlation between grain protein and oil content suggests that both can be enhanced simultaneously [24]. To address the negative correlation between starch content and protein and oil contents, biotechnological methods can be utilized for genetic improvement.

### 3.2. Function Prediction Analysis of Candidate Genes

This study predicted the nine candidate genes most likely associated with this trait. GRMZM2G052213 encodes S-phase kinase-associated protein 1 (SKP1). In Arabidopsis, the F-box protein SKP1 regulates late seed maturation and viability by interacting with chaperone 31 (SKIP31). SKIP31 is primarily expressed in seeds and interacts with jasmonic acid ZIM domain (JAZ) proteins, contributing to the biosynthetic accumulation of abundant proteins, protective metabolites, storage compounds, and abscisic acid during late embryonic development [25]. The plant SKP1 protein is a subunit of the SCF complex E3 ligase and regulates several plant hormone signaling pathways through protein degradation [26]. GRMZM2G301122 encodes glycosyltransferase family 1 (GT1), a protein that plays a crucial role in the biosynthesis of plant secondary metabolites. In maize, GT1 is involved in various biological processes throughout the growth and development stages [27]. GRMZM2G063316 encodes a SET domain protein, the functions and regulation of which involve multiple mechanisms, including protein interactions through both intramolecular and intermolecular associations. These proteins play significant roles in plant developmental processes, such as controlling flowering time and embryonic development [28]. GRMZM2G003411 encodes the G family of ABC transporters. A large number of ABC transporters exist in plants, and these proteins are involved in processes that significantly impact plant adaptability. In Arabidopsis, these transporters contribute to diverse processes, including pathogen response, diffusion barrier formation, and phytohormone transport [29]. GRMZM2G039922 encodes receptor-like protein kinase At1g30570. The At1g promoter has important applications in the growth and biomass improvement of woody plants and, potentially, other plant species [30]. GRMZM2G136910 encodes abscisic acid stress ripening 1 (ASR1). Abscisic acid (ABA), a carotenoid synthesized from carotenoid cleavage, helps maize adapt to various environments by regulating growth, development, defense, and nutrient distribution [31]. GRMZM5G830983 encodes UDP-arabinopyranose, which serves as a key precursor for all glycosylation reactions and is necessary for the synthesis of oligosaccharides and polysaccharides, as well as for protein and lipid glycosylation. Among all nucleotide sugars, UDP-sugar is particularly important for biomass production in nature. UDP-Glc not only functions as a precursor but is also regarded as a signaling molecule that plays a potential role in plant growth and development [32]. GRMZM2G422083 encodes an MYB superfamily protein; these are critical factors in the regulatory networks that govern development, metabolism, and responses to both biotic and abiotic stresses [33]. GRMZM2G089484 encodes mitogen-activated protein kinase 6 (MAPK6), a protein that regulates corn kernel weight. ZmMAPK6 is predominantly located in the nucleus and cytoplasm, and it is widely distributed across various tissues, exhibiting expression during kernel development. Additionally, it influences starch granules, starch content, protein content, and kernel grouting properties [34].

### 3.3. Combination of GWAS and Meta-QTL to Analyze the Genetic Basis of Quality Traits

Although numerous QTL associated with quality-related traits have been identified in maize, their application in genetic improvement programs remains limited, primarily due to their modest effects and environmental variability. In breeding programs that utilize different populations, it is common for the QTL identified in one mapping population to lack validity in others. Reanalyzing identified QTL through QTL meta-analysis presents a promising approach to integrate these findings and predict stable and robust MQTL. This meta-analysis method amalgamates data from various QTL mapping studies conducted across different environments and genetic backgrounds, allowing for the identification of stable, major, and reliable MQTL with reduced confidence intervals [35]. Consequently, the primary objective of MQTL analysis is to pinpoint stable QTL within the genome that can be effectively employed in breeding programs via marker-assisted selection [13].

Among the 40 MQTL identified in this study, 22 MQTL were associated with functional genes and the candidate genes related to quality traits previously reported in corn. This finding suggests that meta-analysis is an effective and feasible approach for mining the genes associated with quality traits in corn. The three significant SNPs identified through GWAS in this study all resided within the MQTL interval, while eight additional significant SNPs were located within 2 Mb of the MQTL. The presence of significant SNPs within or near MQTL regions corroborates the existence of quality candidate genes in these genomic areas. Consequently, these SNPs and intervals, validated by both GWAS and MQTL analyses, should be prioritized for the identification of candidate genes that regulate maize quality traits. The mutual validation of GWAS and MQTL enhances the reliability of the findings in this study. The absence of validation for the remaining MQTL with significant SNPs may be attributed to the genetic diversity present in the study population. This observation also suggests that certain QTL or SNPs identified in mapping studies may be specific to certain breeding lines and are not universally applicable across other accessions [36]. Therefore, the results of this study, in conjunction with GWAS and meta-analysis, will offer valuable insights for more accurately localizing quality-related genes and elucidating the associated molecular mechanisms.

Genome-wide association studies (GWAS) and Meta-QTL analyses primarily focus on the genetic basis of corn quality traits from a genomics perspective, which presents certain limitations. Quality-related traits are complex and are closely linked to population structure. The application of GWAS and QTL to identify the regulatory genes associated with target traits is significantly influenced by population structure, and these methods often lack sufficient detection capability for identifying the quantitative traits governed by specific micro-effect polygenes [37]. Consequently, accurately identifying micro-effect sites or genes poses a challenge, leading to potential false positive or false negative results during the detection process, which can compromise the validity of experimental findings. Furthermore, a comprehensive analysis that integrates genomics, transcriptomics, proteomics, and metabolomics is necessary. Multi-omics analysis represents an effective approach for identifying the key genes associated with various traits and serves as a robust strategy for exploring the genetic basis of these traits, as it compensates for the limitations inherent in each individual method [38,39]. By integrating information across multiple omics levels, multi-omics approaches provide more substantial evidence for elucidating biological mechanisms.

## 4. Materials and Methods

### 4.1. Materials and Experimental Design

This study analyzed 260 corn inbred lines that exhibited superior adaptability among 368 materials provided by the Beijing Academy of Agricultural Sciences, China. In 2020, the corn breeding experimental field located at Huangyang Experimental Station in Wuwei City, Gansu Province, China (37.97° N, 102.63° E) was utilized for planting. A completely randomized block design was implemented, consisting of three replications. Each maize inbred line was planted in two rows, each measuring 4 m in length, with a plant spacing of 0.25 m and a row spacing of 0.4 m. Field management practices adhered to local agricultural production standards.

### 4.2. Determination of Phenotypic Traits

After harvesting, the kernel-related quality traits were measured. The protein content (PC), oil content (OC), and starch content (SC) of corn kernels were measured using the NIRS DA 1650™ (Foss, Hilleroed, Denmark). All results were reported on a dry basis (%), with each sample being repeated three times, and the average value utilized for statistical analysis.

### 4.3. Phenotypic Data Analysis

Descriptive statistical analysis and correlation analysis of the phenotypic data were conducted using IBM SPSS Statistics 21 and Origin 2022 software, and the corresponding charts were generated.

### 4.4. Genome-Wide Association Study

Genome-wide association analysis was performed on 558,529 SNP sites with a minimum allele frequency of ≥0.05, distributed across the entire maize genome. Genotype data can be accessed on the website (http://www.Maizego.org/Resources.html, accessed on 1 June 2024) [10]. Quality control of the genes was performed, and the GWAS analysis was executed using the mixed linear model (MLM) approach in Tassel 5.0 software. The optimal model was selected based on its quantile–quantile (QQ) scatter plot to conduct GWAS analysis for each trait. The Bonferroni threshold was determined using R language, with a significance threshold set at *p* = 1 × 10^−5^ for identifying significant SNPs. QQ plots were generated using the “CMplot” package in R.

### 4.5. Candidate Gene Mining and Functional Analysis

Previous researchers have utilized approximately 560,000 SNPs to assess the degree of linkage disequilibrium (LD) attenuation within this group, revealing that the LD attenuation distance for the associated group is 50 kb (R^2^ = 0.1). Consequently, this study adopted this decay distance as the LD decay distance for the population. This was based on the physical locations of significantly associated SNP markers, both upstream and downstream, of the maize B73 genome sequence RefGen_v3 within a total span of 100 kb. Using resources from the NCBI (https://www.ncbi.nlm.nih.gov/) and MaizeGDB (https://www.maizegdb.org/), we searched for all candidate genes related to each trait and selected the most suitable gene as a candidate, based on its functional annotation. The transcriptome data from various maize tissues were downloaded from the qTeller website (https://qteller.maizegdb.org/, accessed on 1 June 2024). This comprehensive dataset includes 28 tissues and developmental stages [40], such as the embryo (16, 18, 20, 22, 24 DAP), endosperm (12, 16, 18, 20, 22, 24 DAP), whole seed (2, 4, 6, 8, 10, 12, 14, 18, 20, 22, 24 DAP), anthers (R1), cob (R1, V18), silks (R1), and ear primordium (2–4 mm and 6–8 mm). This dataset facilitated tissue-specific expression analysis of the candidate genes.

### 4.6. The Collection of QTL Information Related to Corn Quality Traits

This study utilized the Web of Science (http://www.webofknowledge.com/) and China National Knowledge Infrastructure (https://www.cnki.net/) to gather information on the quantitative trait loci (QTL) associated with corn quality traits. Accessed on June 1, 2024, searches were conducted using keywords such as “corn”, “quality”, “starch”, “oil”, “protein”, “QTL”, and “high-density genetic map”, resulting in the identification of over 20 relevant documents published between 2005 and 2024. Articles that provided QTL intervals and flanking markers were selected for compilation (Appendix A). For experiments with comprehensive genetic maps and QTL information, data were organized according to the formatting requirements of the software. This included all essential parameters, such as QTL name, position, trait, linkage group, LOD (likelihood of odds) value, confidence interval (CI), and phenotypic variance explained (R^2^). Additionally, information from the IBM2 2008 Neighbors map was downloaded from the MaizeGDB website (https://maizegdb.org/data_center/map?id=1140201, accessed on 1 June 2024) to serve as a unified genetic and reference map.

### 4.7. Integration of QTL Information

QTL projection and meta-analysis were conducted using BioMercator 4.2.3 software [41]. The confidence interval (CI) and R^2^ of QTL are two critical parameters. The meta-analysis of QTL primarily relies on the QTL LOD score, R^2^, position, and CI. In cases where the collected QTL data lacks a 95% CI, an inference is made according to the formulas provided by Darvasi and Soller [42], where N represents the size of the original mapping population.
CI = 530/(N × R^2^)(1)
CI = 163/(N × R^2^)(2)

Formula (1) is applicable to backcross and F_2_ mapping populations, while Formula (2) is suitable for RIL mapping populations.

### 4.8. QTL Prediction and Meta-Analysis

The complete genetic map and collected QTL information should be uploaded through genetic data loading, followed by mapping the QTL to the reference map. The genetic linkage map of IBM2 Neighbors integrates the maize high-density molecular marker linkage map (Intermated B73 × Mo17 Map; IBM) with additional molecular marker linkage maps. This comprehensive map comprises 19,111 loci, which include RFLP, SSR, and RAPD markers, as well as genes and sequence probes, extending over a total length of 7898.35 cM (source: https://maizegdb.org/data_center/map, accessed on 10 June 2024; last updated on 10 August 2022, by Marty Sacks). MQTL analysis was conducted on the QTL clusters present on each chromosome. In this analysis, various criteria—AIC (Akaike information criterion), AICc (AIC correction), AIC3 (AIC3 candidate model), BIC (Bayesian information criterion), and AWE (average weight of evidence)—are employed to evaluate all potential QTL combinations. The model exhibiting the lowest AIC value indicates the presence of multiple QTL, with the initial number of QTL for meta-QTL analysis set at a minimum of three [43]. Additionally, the position and confidence interval (CI) at 95% of the QTL are calculated, and flanking markers for the QTL are identified from MaizeGDB. Upon obtaining the MQTL results from the software, the physical location of the corresponding markers on B73 (AGPv 3) of IBM2 Neighbors from 2008 is determined using MaizeGDB. Finally, candidate genes associated with the identified QTL are also retrieved from MaizeGDB.

## 5. Conclusions

In this study, 23 SNPs that significantly correlated with three grain quality traits were identified, along with nine candidate genes predicted to be potentially related to these traits. Through a meta-analysis of grain quality-related traits, 40 meta-QTL (MQTL) and 23 functional and candidate genes associated with quality traits were identified. The genes previously reported in maize were found in 22 MQTL. Three significant SNPs identified through genome-wide association studies (GWAS) were located within these MQTL intervals, while an additional eight significant SNPs were situated close to MQTL (within 2 Mb). The joint analysis of GWAS and meta-QTL enhances the accuracy and reliability of the experimental results and provides a reference for analyzing the genetic basis of SNPs and QTL related to corn quality traits. These findings will offer valuable information for the cloning and breeding of maize quality-related genes.

## Figures and Tables

**Figure 1 plants-13-02730-f001:**
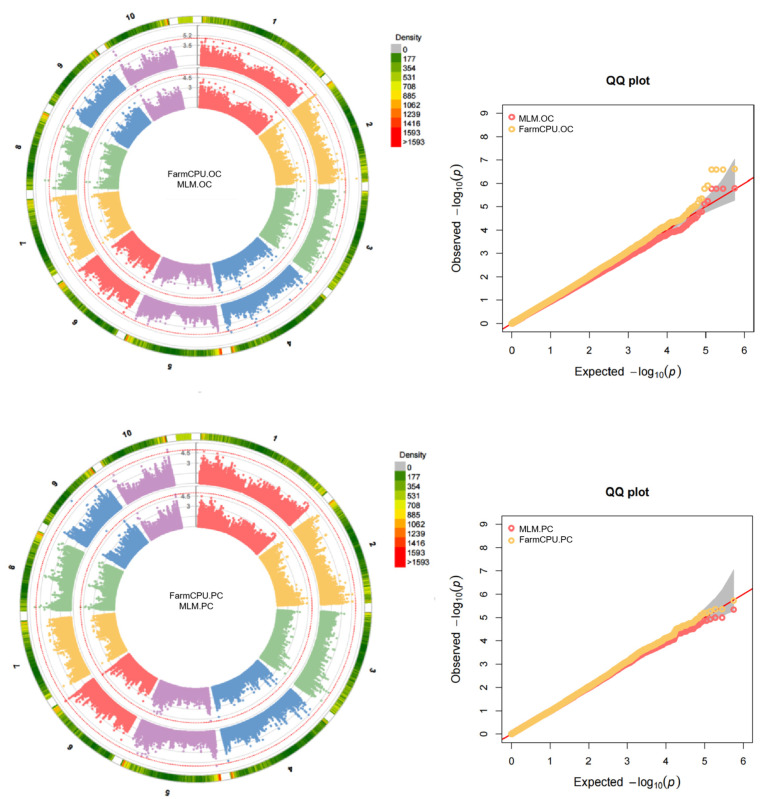
Manhattan plot and QQ plot of the BLUE value of seed quality traits. OC, oil content; PC, protein content; SC, starch content. The red dashed line is the threshold line.

**Figure 2 plants-13-02730-f002:**
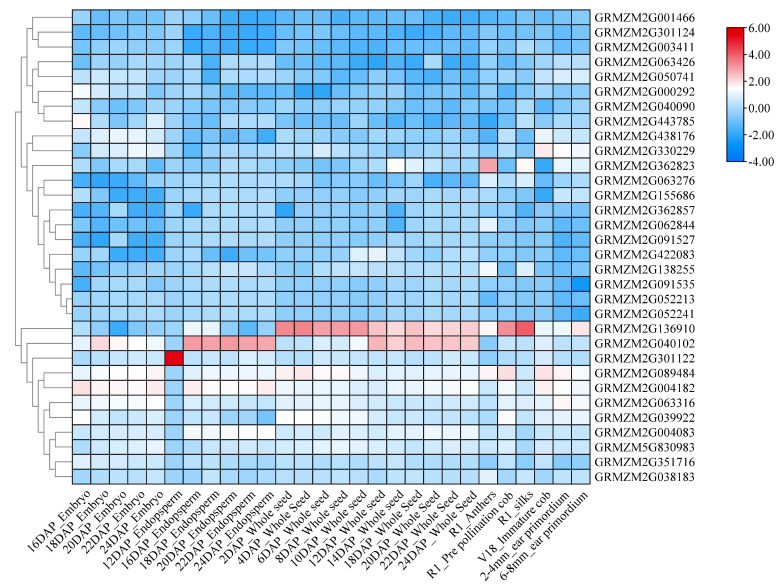
Heatmap of the tissue-specific expression patterns of candidate genes. DAP: Days after pollination, V18: Vegetative stage18, R1: Reproductive 1. Standardized data conversion with log_2_ (FPKM + 1).

**Figure 3 plants-13-02730-f003:**
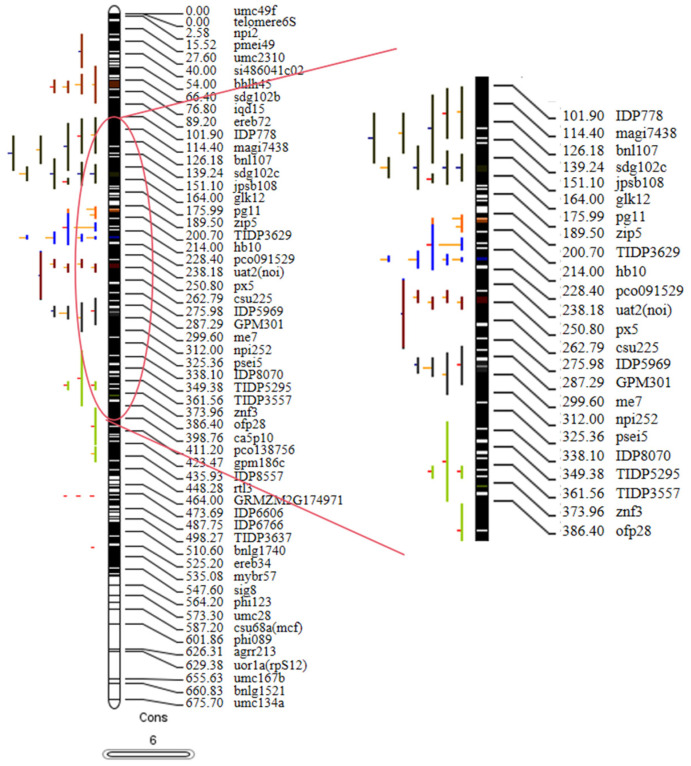
Illustration of the projection and distribution of quantitative trait loci (QTL) and Meta QTL (MQTL) identified for quality traits on chromosome 6. The bars on the left side of the chromosome correspond to QTL associated with quality traits, while the black bars within the chromosome indicate marker density. The colored segments within the chromosome represent MQTL and, on the right side, molecular markers and genetic distances (cM) are displayed.

**Figure 4 plants-13-02730-f004:**
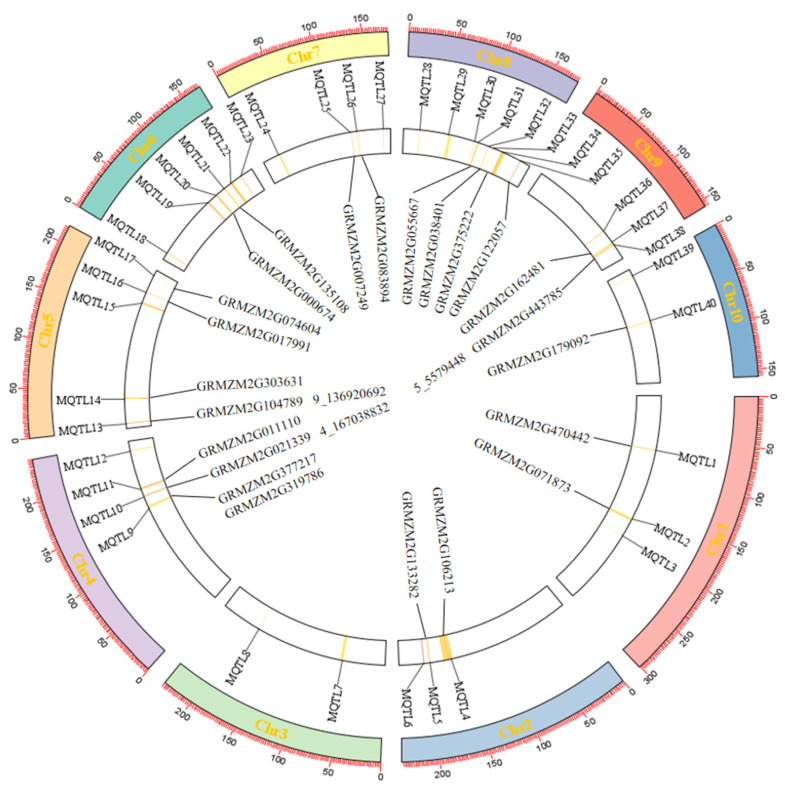
Circos plot illustrating the distribution of MQTL (Meta QTL) and significant SNPs from GWAS studies in maize. From the innermost to the outermost circle: the innermost circle represents the gene density within the MQTL, as well as genes associated with quality; the middle circle depicts the physical map position of the MQTL.

**Table 1 plants-13-02730-t001:** Phenotypic identification of maize inbred lines.

Character	Mean	Range	CV/%	Skewness	Kurtosis
Protein content	11.22 ± 1.14	7.74–14.58	10.00	0.16	0.22
Oil content	3.29 ± 0.72	1.79–4.97	18.00	0.51	0.92
Starch content	61.06 ± 1.83	55.38–66.04	3.00	−0.48	0.21

**Table 2 plants-13-02730-t002:** Candidate gene functional annotation.

Trait	SNPs	Gene	Annotation
OC	2_229132132	GRMZM2G052213	SKP1-like protein 1
OC	10_8849369	GRMZM2G301122	GT1 superfamily protein
OC	4_167038832	GRMZM2G063316	SET domain containing protein
OC	4_231963888	GRMZM2G003411	ABC transporter G family member 39
OC	7_4959916	GRMZM2G039922	Probable receptor-like protein kinase At1g30570
OC	10_8849424	GRMZM2G136910	Abscisic stress protein homolog
SC	9_136920692	GRMZM5G830983	Probable UDP-arabinose 4-epimerase 3-like
SC	10_88685468	GRMZM2G422083	Putative MYB DNA-binding domain superfamily protein
SC	10_88685442	GRMZM2G089484	MAP kinase6

## Data Availability

Data are contained within the article and Appendix A.

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
