# Peer review of "GWAS and Meta-QTL Analysis of Kernel Quality-Related Traits in Maize"

_plants, 2024, doi:10.3390/plants13192730_

Round 1

Reviewer 1 Report

Comments and Suggestions for Authors

Paper with relevant results. Title of the manuscript is suitable, abstract is enough informative. Methods are properly described. Plant material description is not enough detailed. Conclusions are based on the results.  Results and conclusions are clearly presented. Introduction is well informative.

Suggestions:

Line 26. »L.« should be not in italic.

Lines 27-29.  Consider use for food. Given content numbers are approximate. Give better literature citations here.

Lines 297-298. More detailed data on material are needed here. For example: how inbred lines were obtained, when and where material was grown etc.

Line 302. Give detail data on FOSS (address).

Reviewer 2 Report

Comments and Suggestions for Authors

This is a really interesting  study involving 260 corn inbred lines as research materials.  A genome-wide association study (GWAS) was conducted using the best linear unbiased estimator (BLUE) for quality traits, resulting in the identification of 23 significant single nucleotide polymorphisms (SNPs). Additionally, nine genes associated with grain quality traits were identified through gene function annotation and prediction.

The authors complemented the experimental work by compiling 27 documents (a total of 697 quantitative trait loci related to quality traits) and performing a meta-QTL analysis that revealed 40 meta-QTL associated with these traits. Among these, 19 functional genes and reported candidate genes related to quality traits were detected. 

According to investigators , 3 significant SNPs identified by GWAS were located within the intervals of these QTL, while the remaining eight significant SNPs were situated within 2 Mb of the QTL.

The proposal is very interesting but has very important shortcomings, mainly in materials and methods. so I must recommend that it be reconsidered after a major revision.

Below there I added my commentaries.

. Materials and Methods

4.1 Materials and Experimental Design …

4.2 Determination of Phenotypic Traits

It is not established at all how the lines were grown (in what geographical position, the type of soil, the climate and what type of distribution of the material and how it was harvested). If the data came from another work, I can't find the citation either.

4 Genome-Wide Association Study 308 Genome-wide association analysis was performed on 558,529 SNP sites with a mini- 309 mum allele frequency of ≥0.05, distributed across the entire maize genome. Genotype data 310 can be accessed on the website (http://www.Maizego.org/Resources.html) (n = 368).

I accessed to the website but it contains many publications, it must be specified which was the work where to download the markers.

4.4 Genome-Wide Association Study

The method and software of the GWAS are not mentioned.

4.5. Candidate gene mining and functional analysis  

it is repeated 2 times, the paragraphs with almost the same.

2.3. Expression Analysis of Candidate Genes

The expression characteristics of the candidate genes identified at the main stages 101 and in various tissues were analyzed using the public database qTeller. (It is not mentioned in materials and methods, neither the citation)

Using resources from NCBI (https://www.ncbi.nlm.nih.gov/) and MaizeGDB 335 (https://www.maizegdb.org/), we will search for all candidate genes related to each trait 336 and select the most suitable gene as a candidate based on its functional annotation. Additionally, we will download RNA-seq data for the B73 inbred line from panicle development-related tissue sites available at MaizeGDB to perform tissue-specific expression  analysis of the candidate genes. (It is not clear at all)

Comments on the Quality of English Language

y, Zhang et al. [6]utilized (space) three populations 43 across three environments to identify 38  QTLs associated with corn grain quality traits

152  The distribution of the 40 identified  QTLs  across the 10 maize chromosomes is uneven:

349 ..., trait, linkage group, LOD (Likslodd) value, CI, phenotypic ...

387 .MQTL analysis is conducted on the QTL clusters present on each chromosome. (add space)

Round 2

Reviewer 2 Report

Comments and Suggestions for Authors

The authors have responded to all my comments. I recommend acceptance of the manuscript.